# Development, Feasibility, Impact and Acceptability of a Community Pharmacy-Based Diabetes Care Plan in a Low–Middle-Income Country

**DOI:** 10.3390/pharmacy11040109

**Published:** 2023-06-26

**Authors:** Fatima S. Abdulhakeem Ikolaba, Ellen I. Schafheutle, Douglas Steinke

**Affiliations:** Division of Pharmacy and Optometry, School of Health Sciences, The University of Manchester, Manchester M13 9PT, UK; ellen.schafheutle@manchester.ac.uk (E.I.S.); douglas.steinke@manchester.ac.uk (D.S.)

**Keywords:** person-centred care, diabetes, community pharmacy, goal setting, motivational interviewing, pharmacy practice research, Nigeria, Medical Research Council framework

## Abstract

Informed by existing research, mostly from high-income countries, this study aimed to develop and test the feasibility of a community pharmacy person-centred goal-setting intervention for people living with type 2 diabetes in a low–middle-income country—Nigeria. The Medical Research Council (MRC) guidance for developing complex interventions framed the intervention development. Patients participated in monthly community pharmacist consultations over six months. Self-reported and clinical outcome measures were collected at baseline and study completion and analysed in STATA V.14. Twenty pharmacists in 20 pharmacies completed the research and enrolled 104 patients. Of these, 89 patients had complete study data, and 70 patients also completed a post-study evaluation questionnaire. In addition, 15 patients and 10 pharmacists were interviewed. All outcome measures showed statistically significant improvements (*p* < 0.05). Clinical outcomes (BMI, waist circumference, and fasting plasma glucose) improved significantly. Mean patient activation measure (PAM©), quality of life (EQ-VAS©), and medication adherence improved from baseline to study completion. Eighty-eight per cent of questionnaire respondents were satisfied with the service. Interviews indicated care plan acceptability, patient satisfaction, empowerment, and service enthusiasm. Identified barriers to the consultations included time and technology. This study developed a feasible, effective, well-perceived community pharmacy diabetes care plan in Nigeria.

## 1. Introduction

Diabetes is an increasing global health burden, causing significant morbidity, mortality, and resource impact. The development and progression of type 2 diabetes, the most common form [1], and its related complications are linked with modifiable risk factors such as unhealthy dietary habits, obesity, and poor lifestyle [2,3,4]. Type 2 diabetes affects more than four million people in Nigeria [5,6], a low–middle-income country with a population of over 200 million [7]. This figure is predicted to almost double over the coming years [8]. This diabetes threat necessitates innovative approaches to its management [9].

The traditional approach of viewing healthcare professionals as the main decision-makers has changed, recognising the importance of people managing their own conditions [10,11], including diabetes. Hence, there is a need to adopt a more personalised approach [10,11]. The European Association for the Study of Diabetes (EASD) and the American Diabetes Association (ADA) position statements recommend a person-centred approach in diabetes care [12,13]. Person-centred care views the individual in their world, not the disease, and sees patients as active partners in determining their care and needs [11,14]. The shift to the partnership model and person-centeredness is well demonstrated in the ‘House of Care’ model.

The House of Care model “is a coordinated service delivery model that assumes an active role for patients, with collaborative, personalised care planning at its heart” [11]. Personalised care planning is “a conversation, or series of conversations, between a patient and a clinician, in which they jointly agree on goals and actions for managing the patient’s condition” [15]. Personalised care, a proactive approach, is promising and different from the usual reactive care [15,16,17], with patients leading the discussion and agreeing on their own goals [15,16]. Patients with diabetes that are actively involved in their health have reduced medical costs and better health outcomes [18,19]. The steps involved in House of Care planning include goal setting, action planning, documenting, follow up, and reviewing [11].

Goal setting improves patients’ engagement and empowerment [11,15] and affects treatment outcomes positively [15,16]. Goal setting can also increase patient activation, which describes “the knowledge, skills and confidence a person has in managing their own health and healthcare” [20,21] which can be measured reliably with the validated Patient Activation Measure (PAM©) tool [20]. Increased patient activation has been shown to improve health outcomes, reduce healthcare costs, improve adherence, willingness to adopt a healthy lifestyle, and clinical indicators within the normal range [20].

Community pharmacists are accessible healthcare professionals that care for people living with diabetes, providing opportunities for community pharmacists to offer extended care. Innovative community pharmacist services supporting people living with diabetes have been developed and tested in high-income countries (HICs) such as Australia, Canada, Denmark, Italy, the United Kingdom (UK), and the United States (US) [22,23,24,25,26,27,28,29,30,31,32,33] with significant improvements in most measured outcomes. However, comparable quality evidence is lacking in low–middle-income countries (LMICs) [34], including Nigeria. The COVID-19 pandemic impacted healthcare systems globally and resulted in the development of innovative and remote services, including pharmacy services, especially in HICs [35,36]. This study aimed to develop and test the feasibility of an evidence-informed community pharmacy-based care plan delivered remotely to support people with type 2 diabetes in Lagos, Nigeria.

## 2. Materials and Methods

The adapted Medical Research Council (MRC) guidance (Figure 1) for developing complex interventions [37,38] framed the development, feasibility, and evaluation of a person-centred diabetes care plan with multiple components.

### 2.1. Intervention Development and Implementation

The development stage comprised three steps. In step 1, two reviews of the existing published literature were undertaken: (a) a review of pharmacists’ services for people with long-term conditions and (b) a review of community pharmacy-based diabetes care. Review findings [22,26,27,28,31,32,39,40,41] identified practical intervention components for a diabetes care plan (step 2) involving person--centred goal setting and motivational interviewing. The intervention components of the diabetes care plan were identified as pre-intervention strategy, using patient narrative as the starting point, patient education, discussions of self-management support and healthy living, follow-up/monitoring, motivational interviewing, and goal setting. The literature further established the dearth of research in community pharmacy-based diabetes care in LMICs. Other aspects of the care plan delivery and evaluation not informed by the literature review were informed by findings from further developmental work [12,14,42,43,44,45].

Co-designing workshops and meetings were conducted with 19 Nigerian patients, 71 pharmacists, and three physicians (step 3). The stakeholders’ feedback was used to ensure care plan suitability for the Nigerian context and to fine-tune the care plan intervention components. Furthermore, questionnaires and forms were adapted to “lay language”.

The structured care plan was developed as a 6-month, person-led goal-setting intervention, where the patient, not diabetes or medications, was central. The care plan involved trained community pharmacists recruiting people with type 2 diabetes. During remote meetings, the pharmacist identified what the patient perceived as “problems” with managing their type 2 diabetes and supported them in setting personal goals. Subsequently, the patient’s self-set goals were reviewed during regular remote care plan consultations, during which pharmacists provided self-management support using motivational interviewing. 

To participate, a pharmacy had to be licensed in Nigeria [46] and service-oriented with a secure place to hold confidential patient information. To ensure pharmacists understood the intervention and evaluation, they attended a full-day online pre-intervention workshop involving lectures, consultation skills training, role-playing, and hands-on activities. The workshop was facilitated by a consultant endocrinologist, a pharmacist, and representatives from X-PERT Health and Manchester Motivational Interviewing network. A short presentation on how to safely use the glucometer and troubleshoot was also included. All enrolled pharmacists received the presentation slides, leaflets on questionnaire administration and their interpretation, and flyers and videos on how to take accurate measurements.

The workflow of the Diabetes Care Plan is shown in Figure 2.

The red regions in Figure 2 are the evaluation stages of the care plan.

### 2.2. Feasibility

This section reports the feasibility process and acceptability testing. To be included, patients had to be 18 years or older, have type 2 diabetes, which they managed themselves, with adequate mental capacity and ability to read and understand English, and were comfortable using technology. Patients that required further medical diabetes treatment, such as for foot ulcers, or who were pregnant, were excluded. Regarding the sample size, as this is a feasibility study, the number of participants was determined according to the cohort’s membership. The envisaged sample size for pharmacists/pharmacies was 20, and each pharmacy was encouraged to recruit up to six patients, with a planned sample size of 120. The study ran from October 2020 to September 2021. Participants were identified during prescription refills, other diabetes-related visits to the pharmacy and known patients. The pharmacists checked patients’ eligibility and introduced the service to them in a brief conversation. An invitation letter and the other study information were handed to the patient. Only patients who consented to both the service and evaluation were enrolled.

Modification to the care plan: The care plan had originally been designed as face-to-face consultations but was amended to remote delivery due to the COVID-19 pandemic. Face-to-face contact happened when patients received and returned intervention materials and when they had their weight and height measurements scheduled during prescription refills, with precautions taken. Key disease biomarkers (HbA1c, lipid profile, and blood pressure), which had initially been planned, could not be measured. Remote communication was via a secured platform such as WhatsApp video calls, phone calls, voice notes, and text messages, based on patients’ preferences. 

The person-centred care plan is illustrated in Figure 3.

An overview of the consultations performed during the Diabetes Care Plan are given in Figure 4.

### 2.3. Data Collection Methods for Intervention Evaluation

Four sets of data were used to evaluate the intervention’s feasibility, acceptability, and impact/outcomes: patients’ self-reported outcomes using validated questionnaires, all administered pre- and post-intervention; clinical measurements, taken monthly; semi-structured interviews with patients and pharmacists during months 4 to 6; and a post-study questionnaire administered to all patients at study completion. 

Patients were informed and written consent was gained by study pharmacists, including sharing pseudonymised data with the research team. Consenting patients were given forms and questionnaires, and on their receipt, online meetings were booked. A pre-intervention questionnaire collected related patient information. Patients were given glucometers and inelastic tapes to measure their blood glucose and waist circumference monthly. The pharmacists showed the patients how to carry out these measurements accurately and sent them demonstration video links: https://youtu.be/vnC2qqo9XwE (accessed on 8 August 2020) for blood glucose and https://www.youtube.com/watch?v=4MajPk-vp8M (accessed on 8 August 2020) for waist circumference. Patients participated in monthly remote follow-up consultations for six months or, in some cases, more.

Patients’ self-reported outcomes investigated patient activation measured using PAM^®^ [21]; adherence to medication measured using the Morisky, Green, and Levine Adherence Scale (MGL^®^) [48]; and quality of life measured using EQ-5D-5L and EQ-VAS^®^ [49]. Clinical measurements established patients’ plasma glucose level (primary outcome) as well as their weight and height measurements to calculate the body mass index and waist circumference (secondary outcomes). Details on the outcomes and measurement processes are provided in Table 1. The researcher (FI) contacted participants who had consented to arrange semi-structured telephone interviews. The interview topic guides were informed by existing literature [27] and asked about participants’ reasons for enrolling, participation experiences, and barriers and facilitators. Interviews were conducted in English, audio-recorded, and transcribed verbatim. 

The Greater Manchester Community Pharmacy Care Plan (GMCPCP) post-study patient questionnaire [27] was adapted to survey patients’ set goals and success, usefulness of the clinical measurements, attitudes towards the service, experience of managing their diabetes better, and their satisfaction. The questionnaire was handed to patients by their pharmacists during their prescription refill, together with a self-adhesive bag for the return or an online form link via Qualtrics^©^.

### 2.4. Data Sharing and Analysis

Each pharmacist generated a master sheet for their pharmacy through creating a unique patient number (UPN), with only the study pharmacist knowing the link to the patient’s name. These UPNs were used in all study documentation. The pharmacists documented all outputs of the care given in the Patient Care Record Book, kept in a secured cupboard in the pharmacy, and the patient recorded theirs in their diary (patient’s copy). Pseudonymised data were sent to the lead researcher (FI) through password-protected PDF, Word, or Excel documents. The feasibility of intervention was estimated using parameters such as recruitment, retention, attrition, patient engagement with the care plan, intervention acceptability, and likelihood of recommendation. 

On the PAM^®^-13 survey, respondents answered 13 questions and responses were coded on a four-point scale (1 = disagree strongly, 4 = agree strongly) [50]. Responses were converted to a 0–100 scale and categorised into one of four activation levels; a higher PAM score indicates a greater level of activation [50] The MGL measure of adherence was scored according to Morisky and colleagues’ guidance [48], with lower scores indicating better adherence. The EQ-5D-5L^®^ measured health-related quality of life outcomes [51]. 

Interviews were analysed thematically using the six stages outlined by Braun and Clarke [52], aided by NVivo 12. The researcher (FI) coded and analysed the data with regular discussions with her co-authors. Quantitative data were analysed in STATA V 14^®^, (STATA Corp, College Station, TX, USA). Counts and frequencies of categorical data were calculated, and continuous data were reported in means and standard deviation (SD) as well as medians with the range where appropriate. Differences between baseline and study completion were analysed using paired *t*-tests [53]. The chi-square test and McNemar’s test compared categorical variables between different groups and between baseline and follow-up within the same individual (paired categorical data). Post-study questionnaire data were entered into Qualtrics^®^ by FI and exported from Qualtrics^®^ in an Excel^®^ file format. Questionnaire data were analysed using descriptive and inferential analysis in STATA. The significant level (α) for all tests was *p* < 0.05.

## 3. Results

### 3.1. Recruitment and Response Rates

Twenty out of the twenty-seven recruited pharmacies completed the study, recruiting between two and nine patients (median = five). Participation and dropouts are illustrated in Figure 5. Pharmacies were either independently owned or part of a small chain. A total of 104 patients were recruited, and 89 (86%) had complete data for analysis. Seventy (79%) patients completed and returned the post-study questionnaire (PSQ). Telephone (WhatsApp™) interviews were undertaken with ten pharmacists and fifteen patients, lasting 14 to 42 min.

### 3.2. Characteristics of Enrolled Patients

About equal numbers of men (n = 45, 51%) and women (n = 44, 49%) completed the study. Eighty per cent (n = 71) of patients were aged 50 and above, and the mean age was 57. All participating patients had at least a primary school education level, with the majority (n = 64, 72%) having post-secondary education. Owning a business or working in a business establishment was the predominant occupation. Table 2 presents the participants’ characteristics for the feasibility intervention.

### 3.3. Interviewees’ Socio-Demographic Details

The ten pharmacists (eight women) who were interviewed worked in pharmacies that were located in four of the five administrative divisions of Lagos State, and their years of practice ranged from 2 to >30 years, with the majority (60%) practising less than 10 years. Eight out of fifteen interviewed patients were females; their ages ranged from 33 to 75 years, and all had at least a secondary education.

### 3.4. Self-Reported and Clinical Outcome Measures

The mean PAM score was 64 at baseline and 69 at study completion (*p* = 0.01). The proportion of patients with ‘low activation’ (level 1) reduced between baseline and study completion (n = 12 (13%) versus n = 6 (7%)), and ‘highly activated’ patients (level 4) increased between baseline and study completion (n = 26 (29%) to n = 37 (42%). The mean significant difference for EQ-VAS was 7.3 (confidence intervals (CI) of 9.9 and 4.7). (Table 3).There were significant improvements (*p* < 0.05) across all clinical outcome measures from baseline to study end, as shown in Table 3. Using MGL, the proportion of patients with ‘high adherence’ increased between baseline and follow-up (n = 22 (25%) to n = 45 (51%)) (Table 4).

A review of the Patient Care Record Books showed that all participants set at least one goal at baseline and subsequent consultations; most set two to four goals. Similarly, 56 (80%) of the 70 questionnaire respondents set two or more goals. Diabetes control accounted for the highest number (n = 33) of single goals set by participants. This goal was followed by weight loss (n = 15), diet (n = 9), and exercise (n = 9). Diet and exercise were the most common goal combination (n = 72).

### 3.5. Acceptability and Usefulness of the Diabetes Care Plan

The study’s interviews and post-study questionnaires provided further insights regarding care plan acceptability and usefulness. Both interviews are presented together, with respondent type and ID numbers included with each illustrative quote.

#### 3.5.1. Interviews

Interview participants were asked why they joined the service and how they experienced participating in the care plan. One of the main reasons why patients said they joined the service was to broaden their diabetes knowledge. The main reason for interviewed pharmacists was their passion for supporting their patients and helping them understand and manage their diabetes better.


*“I joined the research because I am passionate about patient care. People living with diabetes need special care; some believe that having diabetes is a life sentence. I will acquire more knowledge and still help to debunk that myth about diabetes.”*
Pharmacist 7

Most interviewees (nine pharmacists and all patients) found the care plan rewarding. The interviewees mentioned how community pharmacies could be a base to provide additional support for people with diabetes and how patient empowerment was important in diabetes management. Some patients described how the care plan had provided additional support for their diabetes management, including their mental health.


*“I recommend this service to everyone because it is not all the time that one would want to go to the hospital. […] It’s always good to have a professional that one can talk to, maybe about fear or anxiety or whatever. It helps a lot and is comforting. I am always looking forward to the end of month discussion with the pharmacist because it was beneficial. Reviewing my measurements and goals together and discussing my health has been comforting.”*
Patient 11

On the other hand, one pharmacist mentioned that the care plan delivery was sometimes challenging due to the different schedules of patients and pharmacists.


*“Personally, maybe it’s just for me personally, but timing is a big issue and getting a hold of the participants. Sometimes you plan your time schedule, and they have their own schedule as well, and sometimes they don’t just fall into your plan. And a lot of times, I don’t know if it’s because it’s calls… remote meetings, but they seem to always want it out of hours.”*
Pharmacist 5

Interviewees also identified the benefits of remote consultations, which included inclusivity, especially for home-bound elderly patients, and it being a new mode of patient communication.


*“That (remote consultation) was also a very good one because as much as for us in the pharmacy here, yes, we had some of our clients that were walk-in clients, but we had some geriatric clients also who can’t... don’t come as often as possible. So, the remote consultations came in very handy. So, we were able to communicate with them as much as possible…”*
Pharmacist 2

Some participants also reported innovative and alternative ways of working.


*“Yea, it is the first of its kind, at least to the best of my knowledge within this community. You know, picking up your phone and consulting with a diabetic patient. They were very happy about that, and it is something that one should consider because we are used to this physical consultation, but suddenly, there is this pandemic, and it has made everyone get used to doing some things virtual.”*
Pharmacist 10

Some interviewees reported the care plan’s importance. Patients mentioned how it had impacted them and how it had been comforting and improved their confidence regarding their condition.


*“This programme has impacted me, and I now know that having diabetes is not a life sentence. It has also given me some skills and has helped to improve my confidence in managing my diabetes. Confidence is very important because sometimes, having diabetes can lead to feeling inferior. I don’t struggle with the inferiority anymore. That confidence is there. I am grateful to the pharmacist and the research team.”*
Patient 01

Interviewees were asked about their experience and role in the goal-setting process. Most pharmacists recognised the importance of collaborative care in supporting patients in changing behaviour. Interviewees described how patients were empowered to set their own goals with support.


*“… I allow them to set their goals, to decide their goals by themselves. I would have explained to them where they are, what they can do to get to where they are supposed to be and the consequences of their decisions. I also educate them on how complications can develop. I let them know that the ability to get to where they want is in their own hands, and they are to decide for themselves. More of discussion and putting the ball in their court. Then I follow-up”*
Pharmacist 9

Similarly, several of the patient interviewees reported that the pharmacists identified their concerns and needs regarding their condition and supported them in setting their goals. They also mentioned how the pharmacists discussed and offered them self-monitoring skills. All but one of the interviewees set two or more goals as part of the care plan. The most common goal combinations were diet and exercise.

Some pharmacist interviewees reported patients’ misconceptions about diabetes, such as spiritual beliefs (e.g., curses, healing through going to a place of worship), diabetes not being a long-term condition, or diabetes being a life sentence. The pharmacists supported the patients in addressing these misconceptions.

Some pharmacists said a few patients did not want to go to the hospital for their appointments. The pharmacists reported supporting such patients through discussing the benefits of hospital visits with them and encouraging them to attend hospital appointments regularly. A few pharmacists reported referring their patients to the doctor to improve what they had identified as sub-optimal diabetes management. Identified barriers to the care plan included time and technology.

Finally, pharmacists reported that having access to some clinical measurements they would not normally know helped them to monitor and document and thus better support their patients.

#### 3.5.2. Post-Study Questionnaire (PSQ)

Results of the PSQ showed that a majority (87%) of patients consulted with the same pharmacists in the six remote consultations. Forty-eight patients indicated they also contacted the pharmacists between care plan consultations for ad hoc consultations, with the majority contacting four or more times. Sixty-two patients (89%) scored 4 or 5 to indicate the usefulness of the remote consultations (with 1 = “not at all useful” and 5 = “very useful”). Eighty-eight per cent were satisfied (8 to 10 points) with the care plan (scale: 1 = very dissatisfied to 10 = very satisfied). All respondents answered “yes” to a question asking if participating in the service helped to better manage their condition. All participants responded that they “were likely”/“much more likely” to recommend the care plan to friends and family. Whilst the comparison of mean age and likelihood to recommend the service to family and friends was significant (*p* = 0.048), there were no significant associations between ethnicity and gender with overall satisfaction or likelihood of recommending the service to family and friends.

## 4. Discussion

People living with diabetes have vital roles in self-managing their condition. This study aimed to develop and test the feasibility of an evidence-informed community pharmacy-based care plan to support people with type 2 diabetes. Framed by the MRC framework for complex intervention, an evidence-informed goal-setting intervention was developed for delivery in community pharmacies. The six-month intervention aimed to provide person-centred goal-setting support. 

The care plan feasibility, acceptability, and outcomes/impact were evaluated using validated questionnaires, clinical measurements, semi-structured interviews with patients and pharmacists, and a post-study patient questionnaire. The care plan was successfully implemented in 20 community pharmacies in four (of five) administrative zones of Lagos State, Nigeria. Whilst the care plan had initially been designed to be delivered face-to-face, it had to be amended and delivered remotely following the onset of the COVID-19 pandemic. Eighty-nine patients living with type 2 diabetes ‘attended’ up to six remote pharmacist consultations. Almost equal numbers of males and females with diabetes participated in the care plan, and most participants were above 50 [27,31,33]

Overall, findings demonstrated significant improvements across all outcome measures. Scores for patient activation (PAM©), quality of life (EQ-VAS), and medication adherence (Morisky, Green, and Levine Adherence Scale (MGL)) all improved from baseline to study completion. Clinical outcomes also improved, including plasma glucose, waist circumference, and BMI. The care plan was acceptable, perceived as valuable, and improved patients’ confidence in diabetes management. Significant improvements in most of these outcomes were also reported in previous studies [27,28]. However, the patient groups and study settings differed, as this research focused on people with type 2 diabetes in a LMIC.

Study pharmacists liked delivering the care plan, and patients enjoyed and benefitted from the service. Given the shortages of doctors and inadequate healthcare funding in Nigeria [5], the potential for community pharmacists to provide extended diabetes care is yet to be realised. Previous studies have shown how community pharmacy diabetes care was effective in high-income countries [28,31,54]. The strength of the approach taken for this study is that the draft care plan was co-designed with relevant stakeholders (patients, medical doctors, and pharmacists) to “culturally tweak” the intervention. 

Findings from the study’s interviews and post-study questionnaires suggest that the care plan was perceived as valuable and improved patients’ confidence regarding diabetes management. The research interviews reported some cultural aspects of diabetes, such as patients’ misperceptions about diabetes, spiritual beliefs (e.g., curses, healing through going to a place of worship), and diabetes not being a long-term condition. Some of these misconceptions about diabetes have been identified in previous studies conducted with African Americans [55,56]. These may lower patients’ medicine adherence and negatively influence their diabetes management [55,56]. Some intervention pharmacists described how they spoke to patients about their beliefs about diabetes and provided further support. Monthly online (majority) meetings between the pharmacists and patients maintained the patients’ engagement with the care plan and allowed improvements in goal achievement and follow-up. These findings demonstrate that community pharmacists can proactively support patients with diabetes.

Before the COVID-19 pandemic, studies had shown that remote consultations (telemedicine consultations) were increasing in high-income countries (HICs), and some healthcare systems were encouraging these [57,58,59]. Such remote consultations were highly encouraged and indeed needed at the beginning of the pandemic. Healthcare professionals, including some general practitioners (GPs) and specialists, switched mainly to remote consultations to limit virus spread [60,61]. 

Evidence shows that the COVID-19 pandemic resulted in the development of innovative and remote pharmacy services globally [35,62]. However, while the extension of pharmacists’ roles was often formalised in HICs [36], there is a paucity of reports on expanding pharmacists’ roles in low–middle-income countries (LMICs). As in another study on tele-pharmacy in community pharmacies during the pandemic [62], patients had most of their needs and concerns addressed by community pharmacists. Patients were engaged and looking forward to their monthly discussions with the pharmacists, with some having ad hoc consultations. The remote consultations were perceived as innovative, useful, and inclusive (for elderly, home-bound people). They had a high satisfaction rate among patients and pharmacists, comparable with other video consultations during the COVID-19 pandemic [63]. Hence, part of the novel contribution of this research is the remote delivery of the diabetes care plan in a LMIC. Given that pharmacists had extended their roles from regular to emergency situations in earlier pandemics such as H1N1 influenza and severe acute respiratory syndrome (SARS) [36,64,65,66], their roles in crisis and recovery are vital as the most accessible healthcare professionals and are usually the first point of call. Thus, this research has provided evidence that pharmacists can consult with patients remotely for continuity of patient care.

The House of Care model advocates a radical redesign of services, making patients drive the care process, with collaborative care planning as a key component [11]. Collaborative care planning is an interactive cyclical process, actively involving the patient and healthcare professionals. This work programme showed that community pharmacists used the collaborative care planning part of the diabetes care plan to support patients. Studies have demonstrated that community pharmacists can support people with long-term conditions if collaborative goal setting is incorporated into their practice [27,28]. Pharmacists identified patients’ challenges with their diabetes and concerns that they hoped to change. Pharmacists also provided patients with personalised information based on patients’ needs and offered them self-monitoring skills, including supporting patients with the appropriate use of glucometers and waist measuring tapes. These allowed monitoring, documentation, and interpretation of test results not normally accessible to community pharmacists. 

According to a Cochrane systematic review, an important feature of person-centred care is to explore participants’ experience of the care plan to determine what worked and to explore the patients’ goal attainment through qualitative interviews [15]. Previous care plans either did not explore the views of patients [31,32,33] or focused on exploring the views of patients (intervention recipients) but not the pharmacists (intervention deliverers) [27,41]. This work programme explored both views, offering a different perspective on the service. 

Overall, there is a need to integrate community pharmacy services in Nigeria’s primary care to provide additional patient support as they can proactively support patients due to their accessibility. There is also a need to redesign community pharmacy diabetes services towards person-centred care, including considerations for how best to remunerate such a service.

Study limitations:

Whilst the longitudinal study design was a strength, it was limited due to the lack of a control group. Another limitation was the need to be literate and able to use remote communication technology, which may have excluded more disadvantaged patients. While Lagos state is one of the best-educated states in Nigeria with a high literacy level, the high poverty rate, with 4 in 10 Nigerians below the poverty line [67], may hinder the accessibility and affordability of technology and technological devices. The generalizability of this study is therefore limited, particularly to remote areas.

Thirdly, other key disease biomarkers (i.e., HbA1C, blood pressure, and lipid profile) could not be collected as outcome measures due to the COVID-19 pandemic and the need to minimise face-to-face contact. Moreover, an economic evaluation was not included because this was a feasibility study. However, some previous pharmacy interventions have been shown to be cost effective [28,68].

Lastly, there was a possible selection bias in the interviews, as only those who expressed interest were interviewed. No follow-up on non-participants was done because the research team had no access to patients’ contact details.

## 5. Conclusions

To our knowledge, this is the first comprehensive mixed-methods evaluation of a person-centred goal-setting intervention for people living with diabetes in Lagos, Nigeria. The co-designing ensured partnering and acceptability. This study demonstrated a feasible diabetes care plan in community pharmacies in Nigeria. People living with diabetes who participated in the care plan significantly improved their clinical and self-reported outcomes. The study supports the importance of personalised care to improve people’s empowerment to set goals and outcome measures for their diabetes and wider health. The findings will inform a larger trial.

## Figures and Tables

**Figure 1 pharmacy-11-00109-f001:**
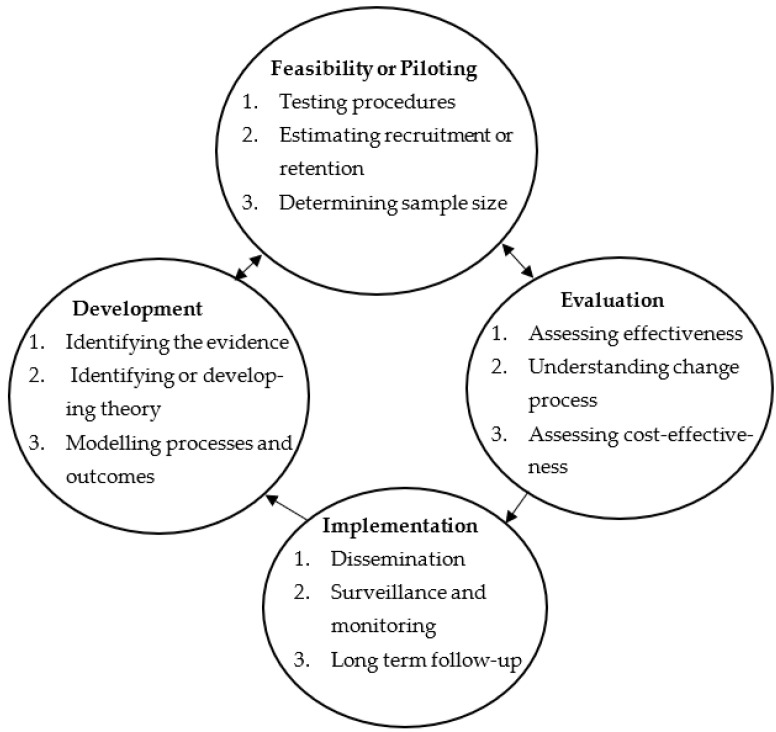
The Medical Research Council model for developing complex interventions [37,38].

**Figure 2 pharmacy-11-00109-f002:**
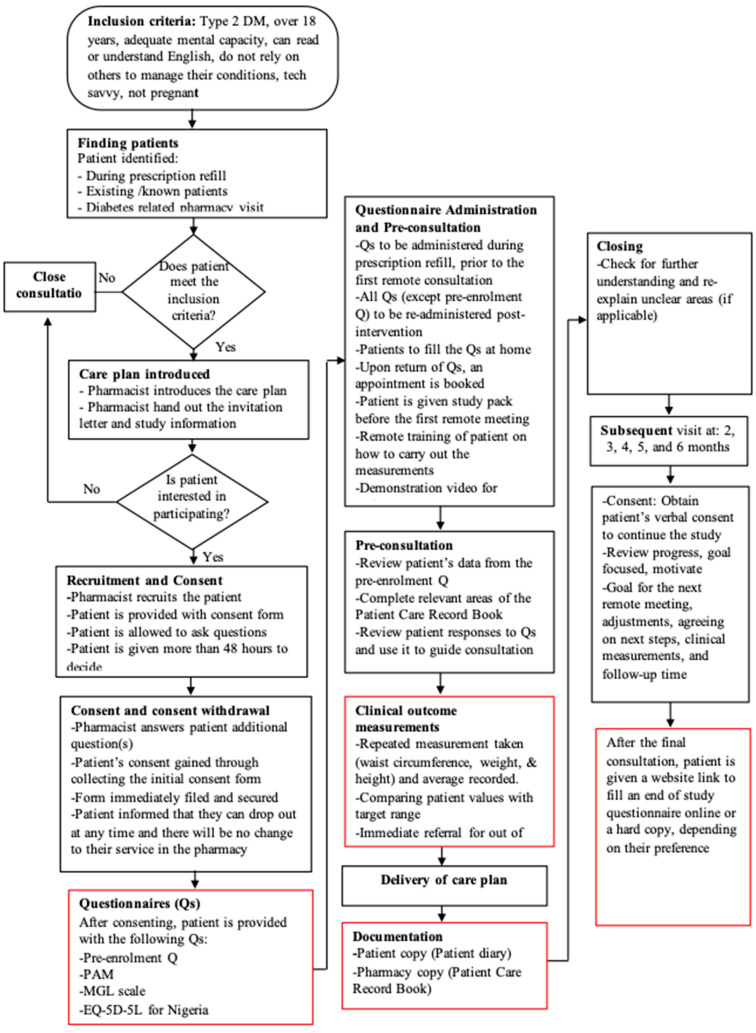
Workflow of the Diabetes Care Plan (adapted from Community Pharmacy Future Pharmacy care plan service [47] and King’s Fund House of Care [11]).

**Figure 3 pharmacy-11-00109-f003:**
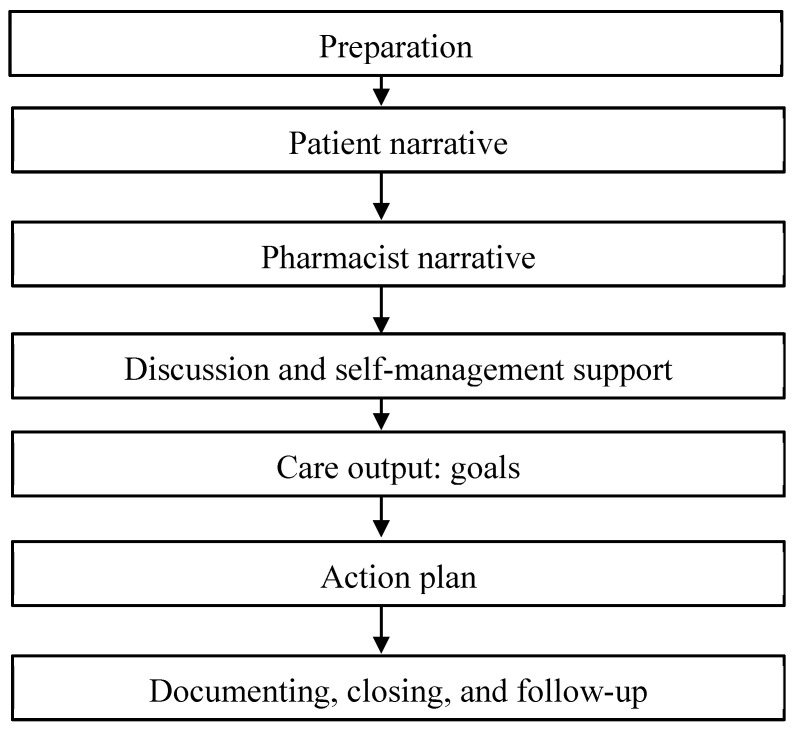
**The person-centred care plan** (adapted from the King’s Fund—building the house of care [11], National Voices [42], and CPF care plan [47]).

**Figure 4 pharmacy-11-00109-f004:**
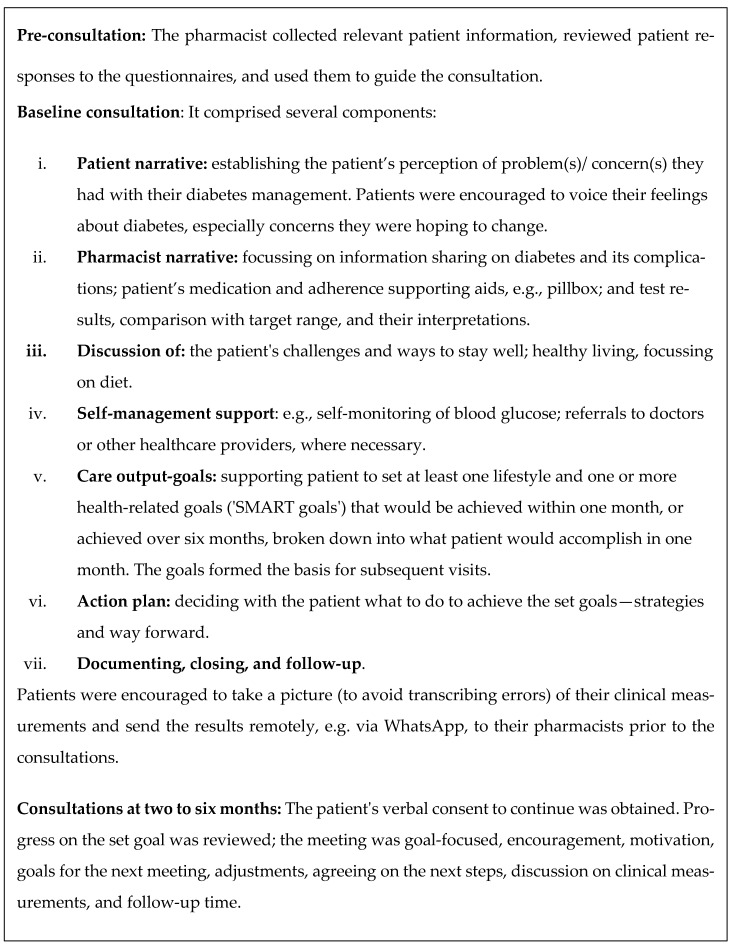
Community Pharmacy-Based Diabetes Care Plan—consultation overview [11,27,47].

**Figure 5 pharmacy-11-00109-f005:**
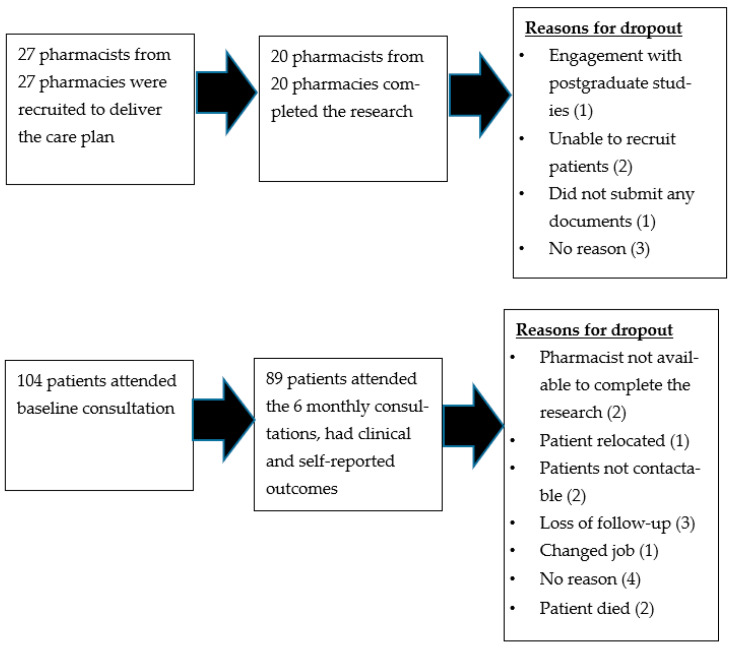
Participation and dropout in the diabetes care plan.

**Table 1 pharmacy-11-00109-t001:** Clinical and self-reported outcome measurements and the process.

Outcome	Measurement Process	Time of Measurement	Statistical Analysis
**Plasma glucose**	Measured with Accucheck Instant^®^ glucometer handed to patients	All consultations	Mean values (and standard deviations) were calculated and comparisons between baseline and study completion were conducted using paired *t*-test.
**Weight and height**	Measured in pharmacy (if patients do not know)	Height once, weight all consultations	Comparison of mean score at baseline and study completion using paired *t*-test
**Waist circumference**	Measured by patients using the provided waist measuring tape	All consultations	Comparison of mean score at baseline and study completion using paired *t*-test
**Patient activation**	PAM-13 [50]: Responses were converted to a scale of 0–100 scale & categorised into 1 of 4 activation levels.	Baseline and on study completion	Paired *t*-test compared the mean PAM score.
**Medicine adherence**	MGL scale [48]: Respondents answered ‘yes’ or ‘no’ to four negative worded questions—one point score for each positive response and 0 point for a “NO” answer.	Baseline and on study completion	Simple proportion.
**Quality of life**	EQ-5D-5L, EQ-VAS [51]. Five statements in EQ-5D-5L. EQVAS records patient’s self-rated health on a vertical visual analogue scale (0–100), endpoints labelled ‘best imaginable health state’ & ‘worst imaginable health state.’	Baseline and on study completion	Comparison of mean score at baseline and study completion using paired t test.

**Table 2 pharmacy-11-00109-t002:** Patient characteristics.

Variable	Frequency (%)
Mean age (SD)	57 (10)
**Age category (years)**	
30–39	3 (3)
40–49	15 (17)
50–59	41 (46)
60–69	17 (19)
70–79	12 (13)
80–89	1 (1)
**Duration of diabetes (years)**	
Less than 5	45 (51)
6–10	9 (10)
11–15	16(18)
16–20	5 (6)
≥21	14 (16)
**Other medical conditions**	
None	37 (42)
Hypertension	42 (47)
COPD/Asthma	1 (1)
Hypertension + Other	6 (7)
Others	3 (3)
**Other providers (apart from doctor) involved in patient’s diabetes care?**	
Yes	64 (72)
No	25 (28)
**Pharmacy where diabetes medication was purchased**	
Hospital Pharmacy	8 (9)
Community Pharmacy	72 (81)
Hospital + Community Pharmacy	9 (10)
**Description of pharmacy use**	
Visit the same pharmacy all the time	39 (44)
Visit variety of pharmacies but one most frequently	45 (51)
Visit variety of pharmacies but none more frequently	2 (2)
Not applicable	3 (3)

**Table 3 pharmacy-11-00109-t003:** EQ-VAS, PAM score, and clinical outcome measures at baseline and on study completion.

Outcome	Population (Baseline, End of Study)	Baseline Mean (SD)	End of Study Mean (SD)	Mean Difference (CI)	*p* Value
EQ-VAS score	89	76.0 (13)	83.0 (12)	7.3 (9.9, 4.7)	<0.001
PAM Score	89	64.1 (16)	69.2 ( 18)	5.1 ( 9.1, 1.2)	0.0116
BMI (kg/m^2^)	89	29.4 (4.9)	28.8 (4.5)	−0.6 (−0.9, −0.3)	<0.001
Waist circumference (cm)	89	99.0 (11.5)	96.7 (9.8)	−2.3 (−3.4, −1.1)	<0.001
Fasting plasma glucose (mmol/L)	61	7.0 (2.3)	6.1 (1.3)	−0.9 (−1.4, −0.4)	<0.001
Phys. Activity (mins/week)	82	95.3 (49.6)	114.8 (47.0)	19.5 (11.5, 27.5)	<0.001

Abbreviations used: BMI, body mass index; Phys. Activity, Physical Activity.

**Table 4 pharmacy-11-00109-t004:** Patients’ adherence at baseline and on study completion.

Outcome	Baseline (n = 89)	End of Study (n = 89)	Statistical Test	*p* Value
High adherence n (%)	22 (25)	45 (51)	n/a	n/a
Medium adherence n (%)	51 (57)	34 (38)	n/a	n/a
Low adherence n (%)	16 (18)	10 (11)	n/a	n/a
Adherence assessment	n/a	n/a	Fisher’s exact	<0.001

## Data Availability

Not applicable.

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
