# Peer review of "Development, Feasibility, Impact and Acceptability of a Community Pharmacy-Based Diabetes Care Plan in a Low–Middle-Income Country"

_pharmacy, 2023, doi:10.3390/pharmacy11040109_

Round 1

Reviewer 1 Report

The topic is very actual and challenging. Real world experiences are very few also in HI countries.

I have just some observations:

- the review of community pharmacy-based diabetes care did not include the Italian studies that do exist

- methods section is too long, plealse shorten it

- a long introduction with a long summary of results in the discussion must be shortened.

English check by a native-English speaker is required

Reviewer 2 Report

This is  a clinically relevant study.  The aim is well described, and the paper is well written.   Some suggestions for minor modification of the paper are:

Line 19/20: suggest rewording the sentence to: "There was a statistically significant improvement in all outcomes..."

Line 77- this is the first time that low middle-income countries is mentioned, so  it would be appropriate to put the abbreviation (LIMCs) here.

 Suggest using tech-savvy rather than techno-savvy in Figure 2

How did the authors decide that  a sample size of 120 would be used? (why 120?)

 Table 1: clarify the plasma glucose statistical entry- It mentions calculating mean values (standard deviation) and then comparing mean score at baseline and completion. It is unclear what the first mean values were from.

Line 358 FI- this should be  "lead author (FI)"

Figure 5: In box Reason for dropout- what does PG stand for? Also, in reasons for dropout, there were 104-89 (=15 dropouts) but only a total of 13 in reasons for dropout. What are the reasons for the other 2?

Suggest reformatting of Table 2

Table 3: would suggest moving Mean(SD) to under Baseline and End of Study

Figure 6 is somewhat confusing. The Y axis is the number of patients but has negative values. Consider a different way to display this data graphically.

Line 588:  suggest changing from significant as p=0.048 to significant (p=0.048)

Line 645: suggest change majorly to mainly

Line 670 - the abbreviation LTC is used, but this is not defined earlier

The study's limitations are described, and the results provide evidence for feasibility, impact and acceptability and form the basis for a larger trial.  The authors should be commended for conducting this clinical research to help the development of pharmacy practice, particularly in the setting of the COVID-19 epidemic.

Reviewer 3 Report

Dear Authors,

congratulations on your valuable work. Here below you can find some suggestions that, in my opinion, could further improve your paper.

1. Pay attention with the authors' list. The first author is called "Firstname Lastname". Please fix this issue.

2. Please carry out a thorough linguistic revision in order to avoid repetitions (e.g. lines 58-59), typos and to make the text more fluid in complex sentences. 

3. Although there is little evidence on the activation of such a project in low-income countries, there are several systematic or scoping reviews in the literature on community pharmacy interventions in these countries. Recently, for example, a scoping review was published specifically in the field of diabetes (https://doi.org/10.1016/j.sapharm.2023.04.124). I suggest an update of the introduction according with this statement.

4. Regarding the MRC model for developing complex interventions, it is not clear to me why you mention it in the methods even though you decided not to use it in your project. I recommend to cite this model in the discussion as a possible further evolution of your study.

5. I think you submitted table 2 as a figure. I suggest to translate it into a table in order to improve the visual quality.

See point n.2 of the "suggestions for autthors"
